# Classification of Acute Myeloid Leukemia by Cell-Free DNA 5-Hydroxymethylcytosine

**DOI:** 10.3390/genes14061180

**Published:** 2023-05-28

**Authors:** Jianming Shao, Shilpan Shah, Siddhartha Ganguly, Youli Zu, Chuan He, Zejuan Li

**Affiliations:** 1Department of Pathology and Genomic Medicine, Houston Methodist Hospital, Houston, TX 77030, USA; 2Neal Cancer Center, Houston Methodist Hospital, Houston, TX 77030, USA; 3Weill Cornell Medical College, New York, NY 10065, USA; 4Houston Methodist Research Institute, Houston, TX 77030, USA; 5Department of Chemistry, Institute for Biophysical Dynamics, The University of Chicago, Chicago, IL 60637, USA; 6Howard Hughes Medical Institute, The University of Chicago, Chicago, IL 60637, USA

**Keywords:** cell-free DNA, 5-hydroxymethylcytosine, acute myeloid leukemia, classification

## Abstract

Epigenetic abnormality is a hallmark of acute myeloid leukemia (AML), and aberrant 5-hydroxymethylcytosine (5hmC) levels are commonly observed in AML patients. As epigenetic subgroups of AML correlate with different clinical outcomes, we investigated whether plasma cell-free DNA (cfDNA) 5hmC could categorize AML patients into subtypes. We profiled the genome-wide landscape of 5hmC in plasma cfDNA from 54 AML patients. Using an unbiased clustering approach, we found that 5hmC levels in genomic regions with a histone mark H3K4me3 classified AML samples into three distinct clusters that were significantly associated with leukemia burden and survival. Cluster 3 showed the highest leukemia burden, the shortest overall survival of patients, and the lowest 5hmC levels in the *TET2* promoter. 5hmC levels in the *TET2* promoter could represent TET2 activity resulting from mutations in DNA demethylation genes and other factors. The novel genes and key signaling pathways associated with aberrant 5hmC patterns could add to our understanding of DNA hydroxymethylation and highlight the potential therapeutic targets in AML. Our results identify a novel 5hmC-based AML classification system and further underscore cfDNA 5hmC as a highly sensitive marker for AML.

## 1. Introduction

Acute myeloid leukemia (AML) is the most common acute leukemia in adults and is characterized by genetic and epigenetic heterogeneity [1]. Despite recent advances in AML research, primary treatment options for newly diagnosed AML patients remain either intensive chemotherapy or less intensive regimens based on hypomethylating agents (HMAs). The five-year survival rate for adult AML patients is only 25% (Cancer.Net). Understanding the molecular basis of AML, including its epigenetic modifications, is crucial for developing new therapeutic approaches for AML patients.

A global decrease in 5-hydroxymethylcytosine (5hmC) levels has been widely observed and closely correlates with poor prognosis in AML [2,3,4,5]. 5hmC is produced as the first oxidative product in the demethylation of 5-methylcytosine (5mC) by ten-eleven-translocation proteins (TET1, TET2, and TET3) [2,3]. Clinically, 5hmC levels are likely a more relevant indicator of disease status than *TET2*, *IDH1,* or *IDH2* mutations [6]. AML patients with somatic mutations or an abnormal expression in DNA-methylation-related genes show significantly reduced 5hmC levels that are relative to other AML patients [4,5,7]. HMAs target DNA methyltransferases, resulting in global hypomethylation and increased 5hmC levels [8,9,10,11]. Given its higher abundance relative to gene mutations [2,12,13], 5hmC is a more sensitive marker for cancer detection.

DNA methylation status can be used to classify AML patients by genetic features and clinical outcomes [14,15,16]. As the first intermediate product of DNA demethylation, 5hmC modification is highly tissue-specific and predominantly stable [2]. Because 5hmC levels are ∼10 to 100-fold lower than 5mC [17], sequencing costs for 5hmC are comparatively inexpensive. The 5hmC-based classification is a promising alternative, as 5hmC is an active DNA demethylation mark that more closely correlates with gene expression compared to DNA methylation [18,19]. The extent to which DNA hydroxymethylation can be used to categorize AML patients has not yet been investigated.

Most studies of 5hmC in AML report 5hmC at global levels due to current technical limitations [2,3,4,5]. A recently developed and highly sensitive nano-hmC-Seal method facilitates the investigation of 5hmC at the genomic level and has significantly increased our understanding of aberrant 5hmC distributions in AML patients [20,21]. Our previous study demonstrated that the number of differentially hydroxymethylated genes with increased 5hmC levels exceeded those with decreased 5hmC levels and involved more signaling pathways in AML patients [21]. Because 5hmC is enriched primarily in distal regulatory regions, gene bodies, and promoters [2,3,20,21,22], the delineation of 5hmC distribution in specific genomic regions may reveal critical aspects of the biology underlying AML. To this end, we and others have shown that plasma cell-free DNA (cfDNA) 5hmC is an effective, minimally invasive, and highly sensitive marker for detection and prognosis in AML and multiple malignancies [2,3,21,22].

To delineate region-specific 5hmC distribution and assess the role of cfDNA 5hmC in AML, we performed genome-wide 5hmC profiling in plasma cfDNA from 54 AML patients using the highly specific nano-hmC-Seal method. We used an unbiased clustering approach and found that 5hmC levels in regions marked with H3K4me3 classified AML samples into three distinct clusters. These clusters were significantly associated with leukemia burden, overall survival (OS), and 5hmC levels at the *TET2* promoter. The classification of AML by cfDNA 5hmC is a new approach for guiding therapeutic treatment decisions in AML. Our findings identify key genes and signaling pathways that are associated with aberrant 5hmC levels in AML and may have significant implications for therapeutic targeting in AML.

## 2. Materials and Methods

### 2.1. Patient Cohort

The study included blood samples from 54 patients diagnosed with AML (Appendix A) that were collected between 2015 and 2022 at Houston Methodist Hospital. The age of the patients ranged from 20 to 88 years. All patients received chemotherapy, and 31 also received hematopoietic stem cell transplantation. Bone marrow samples were analyzed for myeloblasts using multiparameter flow cytometry and gene mutations using next-generation sequencing panels through standard clinical care. This study was approved by the institutional review board at Houston Methodist Hospital.

### 2.2. DNA Extraction and 5hmC Sequencing

We isolated the plasma by centrifuging whole blood at 1350× *g* for 10 min at 4 °C. Plasma cfDNA was extracted from 1 mL of plasma using the QIAamp Circulating Nucleic Acid Kit (QIAGEN, Germantown, MD, USA) following the manufacturer’s instructions. We assessed cfDNA quantity and quality using the Qubit Fluorometer with dsDNA HS Assay Kit (Thermo Fisher Scientific, Waltham, MA, USA) and Bioanalyzer 2100 with Agilent High Sensitivity Assay Kit (Agilent Technologies, Santa Clara, CA, USA). We constructed the 5hmC library as previously described [21,22]. Briefly, we ligated cfDNA with adaptors before incubation with N3-UDP-azide-glucose and T4 Phage β-glucosyltransferase at 37 °C for 1 h. We then purified the DNA and incubated it with DBCO-PEG4-DBCO at 37 °C for 2 h. The DNA libraries were sequenced using the NextSeq 550 and NovaSeq 6000 instruments (Illumina, San Diego, CA, USA). Sequencing data were processed as previously described [21,22]. Briefly, we first evaluated the raw sequencing read quality with FastQC (https://www.bioinformatics.babraham.ac.uk/projects/fastqc/ (accessed on 1 May 2020). We trimmed adaptors and low-quality reads using Trimmomatic version 0.39. We mapped high-quality reads to the reference genome (GRCh38) using the bowtie2 version 2.4.5 with the end-to-end mode. Reads with a mapping quality score ≥20, insert size <600 bp, up to one ambiguous base, and <3 mismatches were retained. We removed PCR duplicates using the SAMtools version 1.16.1.

### 2.3. Analysis of Genome-Wide 5hmC Distribution

We normalized raw gene read counts using counts per million (CPM), and 29,423 genes remained in reads with CPM ≥ 3 in more than 10% of the samples. We obtained histone ChIP-seq peak data for histone marks of H3K4me1, H3K4me3, H3K27ac, and H3K36me3 in common myeloid progenitor (CD34+) and populations from the Encyclopedia of DNA Elements project. High-quality reads were counted into histone-enriched regions and gene bodies (RefSeq) using featureCounts from Subread package version 2.0.0 [23]. To reduce the data for clustering analysis, we selected the 500 most variable regions for pairwise Pearson’s correlation between the samples. Unsupervised hierarchical clustering heatmaps were generated using the Pearson correlation coefficient with pheatmap v1.0.12 (https://cran.r-project.org/package=pheatmap (accessed on 14 September 2020). Clusters were defined based on the dendrogram of the heatmap.

To identify differentially hydroxymethylated regions (DhMRs) among the clusters, differential hydroxymethylation analysis controlling for the effects of age and sex was performed using the R/Bioconductor package DESeq2 (v1.34.0) [24]. Regions with a false discovery rate (FDR) <0.05 were considered DhMRs. DhMRs were annotated using the R package ChIPseeker v1.30.3 [25]. Gene enrichment and signaling pathway analyses were performed with Ingenuity Pathway Analysis (IPA). A list of differentially expressed genes was exported from the GEPIA database (http://gepia.cancer-pku.cn/detail.php (accessed on 14 July 2021) [26]. Venn diagrams were produced using the R package eulerr (https://cran.r-project.org/package=eulerr (accessed on 8 March 2023)). The ggplot2 R package was used for data visualization.

### 2.4. Statistical Analyses

We performed plotting and statistical tests using R language version 4.1.1. Kaplan–Meier curves were used to display OS, and survival curves were generated using the survminer R package (https://cran.r-project.org/web/packages/survminer/index.html (accessed on 26 July 2021). A log-rank test was used to evaluate the statistical significance of OS between the groups. Multivariate Cox regression analysis was performed using the Survival R package (https://cran.r-project.org/web/packages/survival/index.html (accessed on 26 July 2021). Comparisons between the groups were analyzed using the Wilcoxon Rank Sum test. An FDR value <0.05 was considered significant.

## 3. Results

### 3.1. Classification of AML Samples Using Genome-Wide 5hmC Distribution

To assess 5hmC classification in AML samples, we calculated a pairwise Pearson’s correlation coefficient of 5hmC levels in the 500 most variable regions and performed unsupervised hierarchical clustering in 54 AML samples. As 5hmC is highly enriched in genomic regions associated with active histone marks, including H3K36me3 (in gene bodies), H3K4me3 (in promoters), H3K4me1 and H3K27ac (in *cis*-regulatory enhancer elements) [21], we analyzed 5hmC patterns in regions with each histone mark in AML samples. 5hmC levels in the regions of H3K4me3 showed the best classification performance among the histone marks and were used to classify the samples into three distinct groups, clusters 1, 2, and 3 (Figure 1A; Appendix A). The percentage of myeloblasts was significantly higher in cluster 3 (median 38.0%, range 4.0–85.0%) compared to cluster 1 (median 0.0%, range 0.0–22.0%; *p* = 1.2 × 10^−6^) or cluster 2 (median 0.0%, range 0.0–74.0%; *p* = 2.2 × 10^−4^; Figure 1B). Cluster 2 showed a higher percentage of myeloblasts, but this result was not significant in relation to cluster 1 (*p* = 0.26; Figure 1B).

### 3.2. Characteristics of cfDNA in the Clusters of AML Patients

To evaluate the characteristics of cfDNA within each cluster, we analyzed cfDNA’s quantity and quality. The median cfDNA yield among the 54 AML samples was 50.6 ng/mL plasma (range, 11.7–1959.4 ng/mL; Appendix A). The cfDNA yield was significantly higher in specimens from cluster 3 (*p* = 1.7 *×* 10^−5^*)* and cluster 2 (*p* = 0.0056) compared to specimens from cluster 1 (Appendix A). This indicated a correlation between a greater proportion of myeloblasts and a higher cfDNA yield. The median cfDNA fragment size was 174 bp (range, 163–193 bp; Appendix A). No significant difference was found in cfDNA fragment sizes among the three clusters (Appendix A), suggesting that cfDNA quality was not affected by the leukemia burden.

### 3.3. 5-hmC Levels in DNA Demethylation Genes

As the 5hmC level in the H3K4me3 region (promoter) was closely correlated with gene expression [27], and we analyzed the 5hmC levels in this region in DNA demethylation genes, including *TET2*, *IDH1*, *IDH2*, and *WT1* in all the samples. Compared to cluster 1, samples in cluster 2 (*p* = 0.012) and cluster 3 (*p* = 3.3 × 10^−5^) had significantly lower 5hmC levels in the *TET2* promoter (Figure 2A). Interestingly, 8 out of the 12 samples with a mutation in *TET2*, *IDH1*, *IDH2*, or *WT1* were segregated in cluster 3, and only two samples with such a mutation were in clusters 1 and 2, respectively (Figure 2A). The samples with a mutation in DNA methylation genes were also associated with significantly lower 5hmC levels in the *TET2* promoter compared to the samples in cluster 1 (*p* = 0.00038; Figure 2A). In addition, we observed a significant increase in 5hmC levels in the H3K4me3 region of *WT1* in cluster 3 compared to cluster 1 (*p* = 1.3 × 10^−8^) or 2 (*p* = 2.1 × 10^−9^; Figure 2B). No 5hmC level changes in the H3K4me3 region of *IDH1* and *IDH2* were observed (Appendix A). These results demonstrate that 5hmC levels in the *TET2* promoter could be significantly associated with 5hmC clusters and mutations in DNA demethylation genes, suggesting that 5hmC levels in the *TET2* promoter might represent TET2 activity.

### 3.4. Association of 5hmC Clusters with OS in AML Patients

To investigate if the 5hmC clusters were associated with AML prognosis, we analyzed the OS of AML patients in each cluster. The OS for patients in clusters 1 and 2 were comparable [*p* = 0.97; hazard ratio (HR) 0.97; 95% confidence interval (CI): 0.2–6.38; Figure 3]. Patients in cluster 3 showed a significantly shorter OS compared to cluster 1 (*p* = 0.0063; HR 6.2; 95% CI: 1.4–27.9) and cluster 2 (*p* = 0.00078; HR 7.1; 95% CI: 1.9–25.8; Figure 3). The 12-month OS rate was 41.5% in cluster 3 vs. 84.0% in cluster 1 and 88.9% in cluster 2. Multivariate Cox regression analysis showed that the prognostic value of cluster 3 was independent of the myeloblast count in bone marrow (HR 7.8; 95% CI: 1.1–53.3; *p* = 0.03; Appendix A).

### 3.5. Genomic Regions with Aberrant 5hmC Levels in AML

Identifying common and specific genes in each cluster may reveal critical genes for leukemia burden and leukemia development. Because most AML samples in cluster 1 were in complete remission, we used this cluster as a control and compared 5hmC levels in the H3K4me3 regions in clusters 2 and 3 to cluster 1. The total number of DhMRs was 2535 in cluster 2 and 3372 in cluster 3, and 1089 DhMRs were shared by both (Figure 4). More DhMRs had reduced 5hmC levels relative to those with increased 5hmC levels in both cluster 2 (1893 vs. 642) and cluster 3 (1852 vs. 1520; Appendix A). Among the significantly changed DhMR-linked genes, the majority (1644 in cluster 2 and 2095 in cluster 3) were genes that had not undergone previous extensive study in AML according to the IPA database (Appendix A). The remaining genes had been previously reported in AML, including *CDK6*, *ARHGAP26*, and *ETV6*, and were shared by both clusters. *MAP2K2*, *NRIP1*, and *LYL1* were only present in cluster 2, while *ZMYND8*, *WT1*, *PBX1* were only present in cluster 3 (Figure 4; Appendix A).

To assess the impact of 5hmC alterations on gene expression, we correlated DhMRs with gene expression levels in AML patients using data from the Gene Expression Profiling Interactive Analysis (GEPIA) database. In cluster 2, of the 1506 DhMR-related genes presented in the GEPIA database, 883 genes (58.6%) exhibited differential expression at the mRNA level (Figure 4). Similarly, in cluster 3, of the 1920 DhMR-related genes present in the GEPIA database, 1119 genes (58.3%) were differentially expressed (Figure 4; Appendix A). These findings suggest that the DhMRs identified in clusters 2 and 3 were associated with gene expression.

### 3.6. Signaling Pathways with Aberrant 5hmC Levels in AML

To determine the functional impact of aberrant 5hmC distributions, we analyzed the signaling pathways in DhMR-linked genes in clusters 2 and 3. A total of 151 pathways (86.3% of cluster 2 and 61.1% of cluster 3) were shared by both clusters, indicating these were important pathways for leukemogenesis (Figure 5A; Appendix A). Many of these pathways were associated with cell proliferation, such as molecular mechanisms of cancer, protein kinase A signaling, and G-protein coupled receptor signaling (Figure 5B). The pathways independent of each cluster could be associated with leukemia burden or distinct molecular mechanisms. Some of the independent pathways in cluster 2 were associated with targeted treatment, such as peroxisome proliferator-activated receptors (PPAR), WNT/β-catenin, and iNOS signaling pathways (Figure 5C). In cluster 3, the independent pathways were related to cell survival and proliferation, such as mTOR, ERK/MAPK, and insulin receptor signaling pathways, which is consistent with the adverse prognosis in the patients of cluster 3 (Figure 5D).

## 4. Discussion

Previous studies have revealed that DNA methylation patterns can segregate AML patients into subtypes associated with diagnosis and prognosis [14,15,16]. As an intermediate product of DNA demethylation, the 5hmC level provides a stronger positive correlation with the gene expression compared to the 5mC level [19]. We investigated whether plasma cfDNA 5hmC patterns in AML could classify AML patients into distinct clusters. We found that 5hmC levels in genomic regions with the H3K4me3 histone mark outperformed genomic regions with other histone marks (H3K4me1, H3K27ac, and H3K36me3) for AML classification. Using 5hmC levels in H3K4me3 regions, we identified clusters that were significantly associated with leukemia burden and survival. Our results suggest that 5hmC distributions in H3K4me3 regions may closely relate to AML development and disease progression. These findings also support our previous report that plasma cfDNA 5hmC is highly sensitive for AML detection and prognosis [21].

Our results indicated that the 5hmC level in the *TET2* promoter was more relevant than mutations in DNA demethylation genes for assessing TET2 activity. In addition to AML samples with a mutation in *TET2*, samples in cluster 3 showed enriched *IDH1*, *IDH2*, or *WT1* and were significantly associated with reduced 5hmC levels in the *TET2* promoter. In AML, reduced 5hmC levels were observed in patients with or without a mutation in epigenetic modifiers, including *TET2*, *IDH1*, *IDH2*, and *WT1*, suggesting that other genes or posttranslational changes also regulated 5hmC levels [6,28,29,30,31,32,33]. Our findings indicate that low 5hmC levels in the *TET2* promoter likely represent reduced TET2 activity resulting from mutations in DNA demethylation genes and other factors. From a clinical perspective, 5hmC analysis is easier to assess relative to gene expression or other analytes. Compared to the assessment of simple mutations in *TET2*, *IDH1*, *IDH2*, and *WT1,* 5hmC analysis is a promising new approach for guiding therapeutic management. For example, patients with decreased TET2 activity may benefit from treatment with a hypomethylating agent and vitamin C (ascorbate) [28,32,34].

Interestingly, we found increased 5hmC levels in the *WT1* promoter in AML samples without a mutation in *TET2*, *IDH1*, *IDH2*, or *WT1*. WT1 is a transcription factor that recruits *TET2* to DNA, enabling promoter demethylation [35]. Loss-of-function mutations in *WT1* result in a global reduction in 5hmC levels comparable to that seen in AML patients with *TET2*, *IDH1*, and *IDH2* mutations [35,36]. However, *WT1* is overexpressed in the majority of AML patients and is considered an oncogene [36]. Our findings indicated that the overexpression of *WT1* in most AML samples was consistent with previous reports [36]. This finding further suggested multiple functions for *WT1* in AML.

Consistent with a global reduction in 5hmC levels in AML, we found more H3K4me3-associated DhMRs with reduced 5hmC levels compared to increased 5hmC levels. H3K4me3 involves transcriptional initiation and transcriptional pause-release and elongation and is closely associated with gene expression [27]. Our findings suggest that there are more genes with reduced expression than increased expression. We observed increased 5hmC levels in H3K4me3 regions in genes relating to cell proliferation, such as *CDK6* [37] and *ZMYND8* [38], and reduced 5hmC levels in H3K4me3 regions in genes that may be related to tumor suppression, such as *ARHGAP26* [39] and *GLI3* [40]. These results indicate that aberrant 5hmC levels result in unbalanced gene expression, which is an important mechanism of leukemogenesis. Because the majority of these genes are not well studied in AML, future studies investigating their association with aberrant 5hmC levels are warranted and may reveal the critical genes underlying AML development to further our current understanding of DNA hydroxymethylation in AML.

Our classification of AML using 5hmC revealed distinct dysfunctional epigenetic pathways related to gene regulation. As the majority of signaling pathways with aberrant 5hmC levels were shared by clusters 2 and 3 and involved in cell proliferation, targeting these pathways may help inhibit leukemia cell proliferation. Signaling pathways that are specific for each cluster may be associated with the leukemia burden and provide potential drug targets for patients with a specific 5hmC pattern. For example, PPAR signaling was significantly enriched in cluster 2. PPARs are critical sensors and regulators of lipids, and their excessive expression can be related to cell growth and survival in several malignancies, including leukemia [41]. PPAR modulators (both agonists and antagonists) have been widely investigated for the regulation of cancer cell proliferation and differentiation [41]. However, studies in AML are limited. Targeting PPAR signaling is a potential therapeutic strategy for AML patients in cluster 2 and warrants further investigation. Collectively, these results provide critical evidence of DNA hydroxymethylation in AML development and highlight the potential of identifying drug target pathways in cancer subtypes.

Using cfDNA 5hmC to classify AML patients has distinct advantages, given its high tissue and cancer specificity, which may mitigate interference from cfDNA derived from non-cancerous cells [2,3]. However, the application of cfDNA 5hmC in AML can be improved in several aspects. Just as differential DNA methylation patterns can classify AML patients into multiple subtypes [14], 5hmC can also classify AML patients with a high leukemia burden into additional subtypes. 5hmC-based classification may also improve the risk stratification of AML patients. Additional studies on larger cohorts of AML patients are needed. In addition, functional studies on the novel genes and signaling pathways identified in this study could be valuable for assessing their roles in AML.

## 5. Conclusions

Here, we have shown how plasma cfDNA 5hmC is a powerful approach for classifying AML patients. Cluster 3 was significantly and independently associated with prognosis in AML. A deeper understanding of how novel genes and critical signaling pathways influence DNA hydroxymethylation in AML could allow the optimization of therapeutic targets for AML patients. The distinct genetic composition of these clusters sheds light on potential therapeutic approaches for AML patients. Our results further underscore the importance and relevance of cfDNA 5hmC as a marker in AML.

## Figures and Tables

**Figure 1 genes-14-01180-f001:**
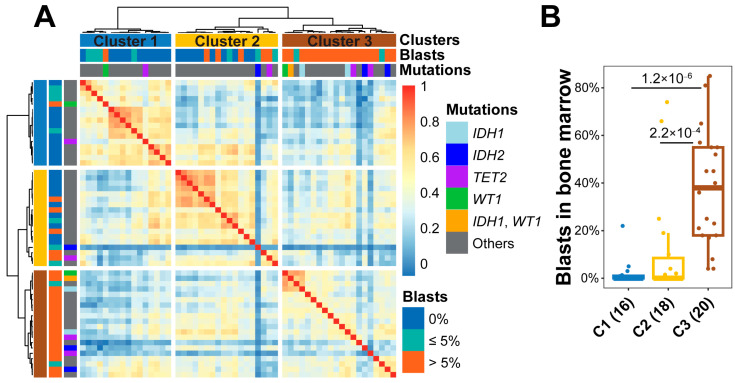
Classification of acute myeloid leukemia patient samples using 5-hydroxymethylcytosine patterns in H3K4me3 regions. (**A**) Unsupervised hierarchical clustering of 54 AML samples using pairwise Pearson’s correlation coefficient of 5-hydroxymethylcytosine (5hmC) levels in the 500 most variable regions marked with H3K4me3. (**B**) Quantity of bone marrow myeloblasts in three clusters. C1, C2, and C3 represent clusters 1, 2, and 3. The number of patients in each cluster is shown in parentheses. *p* values are labeled for clusters with a significant difference in myeloblast percentage. Center line represents median, bounds of box represent 25th and 75th percentiles, and whiskers are Tukey whiskers.

**Figure 2 genes-14-01180-f002:**
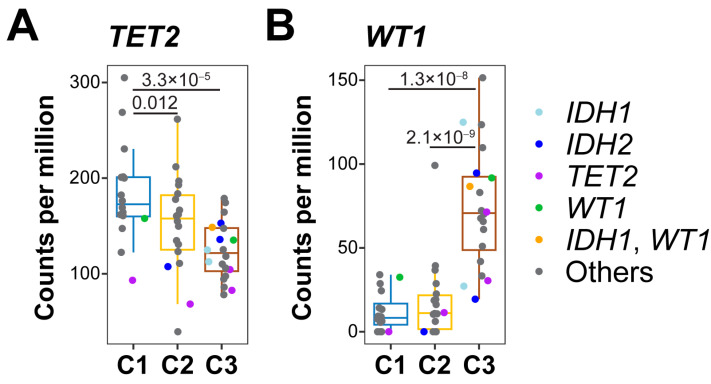
Levels of 5-hydroxymethylcytosine in promoters of *TET2* (**A**) and *WT1* (**B**). 5hmC enrichment in genomic regions marked with H3K4me3 shown for each cluster. C1, C2, and C3 represent clusters 1, 2, and 3. *p* values are labeled for clusters with a significant difference in the percentage of myeloblasts. Center line represents median, bounds of box represent 25th and 75th percentiles, and whiskers are Tukey whiskers.

**Figure 3 genes-14-01180-f003:**
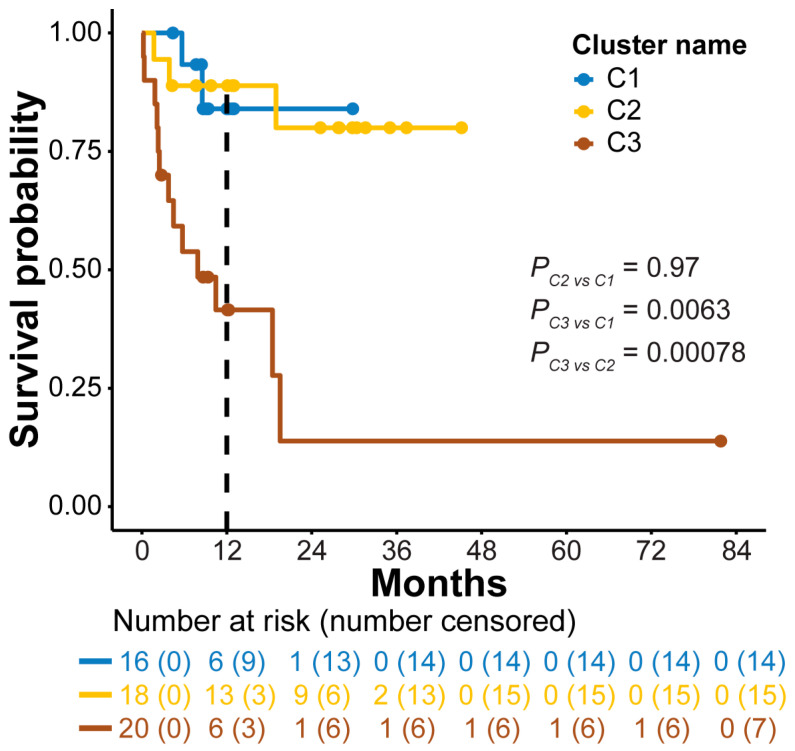
Overall survival of acute myeloid leukemia patients in the 5-hydroxymethylcytosine clusters. Kaplan–Meier analysis of overall survival of AML patients in training set based on wp-scores (*n* = 54). Dots indicate the time being censored. C1, C2, and C3 represent clusters 1, 2, and 3.

**Figure 4 genes-14-01180-f004:**
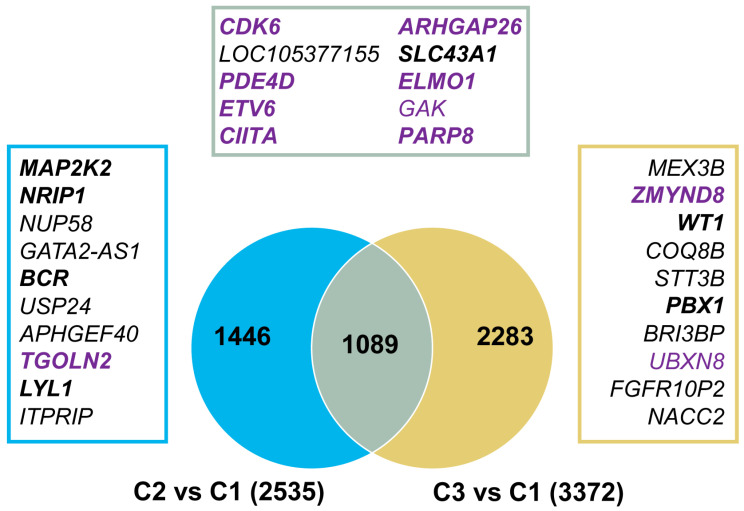
Regions with differential DNA hydroxymethylation in acute myeloid leukemia. Both clusters 2 and 3 were compared with cluster 1. The total number of differentially hydroxymethylated regions (DhMRs) for cluster 2 vs. cluster 1 and cluster 3 vs. cluster 1 are listed in parentheses. The top 10 significant DhMRs shared by both clusters and independent to cluster 2 or cluster 3 are listed in boxes. Genes reported in AML based on BioProfiler in IPA are in bold. Genes with abnormal mRNA expression are purple. Blue represents cluster 2 only. Brown represents cluster 3 only. Grey represents the DhMRs shared by both clusters 2 and 3. C1, C2, and C3 represent clusters 1, 2, and 3.

**Figure 5 genes-14-01180-f005:**
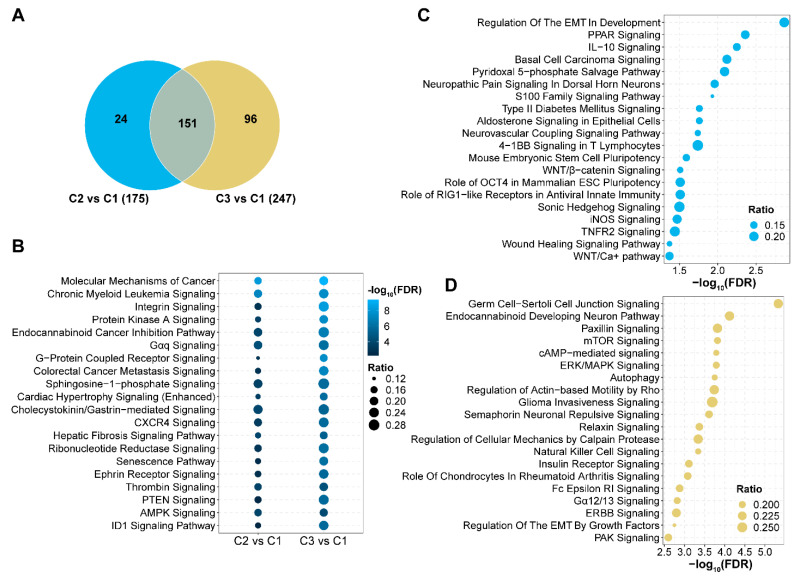
Signaling pathways with aberrant 5hmC levels in acute myeloid leukemia. (**A**) The number of signaling pathways significantly associated with aberrant 5-hydroxymethylcytosine (5hmC) levels in each cluster. Canonical pathway analysis was performed using Ingenuity Pathway Analysis. Both clusters 2 and 3 were compared with cluster 1. The total number of significant signaling pathways for cluster 2 vs. cluster 1 and cluster 3 vs. cluster 1 are listed in parentheses. Blue represents cluster 2 only. Brown represents cluster 3 only. Grey represents the signaling pathways shared by both clusters 2 and 3. C1, C2, and C3 represent clusters 1, 2, and 3. (**B**) Signaling pathways shared by clusters 2 and 3. (**C**) Signaling pathways with aberrant 5hmC levels in cluster 2. (**D**) Signaling pathways with aberrant 5hmC levels in cluster 3. The top 20 most significant signaling pathways are displayed in (**B**–**D**). Ratio indicates the number of genes in each pathway divided by the total number of genes that make up that pathway.

## Data Availability

The raw 5hmC sequencing data supporting the conclusions of this article are available in the National Center for Biotechnology Information (NCBI) Gene Expression Omnibus (GEO) database.

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
