# Peer review of "Classification of Acute Myeloid Leukemia by Cell-Free DNA 5-Hydroxymethylcytosine"

_genes, 2023, doi:10.3390/genes14061180_

Round 1

Reviewer 1 Report

In the manuscript by J. Shao et al. with the title “Classification of Acute Myeloid Leukemia by Cell-Free DNA 5-hydroxymethlcytosine” the authors describe the use of cell free circulating DNA to classify AML using the epigenetic mark 5-hydroxymethlcytosine (5hmC). The letter is the first step of the demethylation of 5-methlycytosine by TET proteins. The authors perform the analysis in a genome wide fashion and then restrict their analysis to different regions of the genome defined by specific markers on histone tails, that preferentially bind promoters (H3K4me3), cis-regulatory regions (H3K27ac, H3K4me) or gene bodies H3K36me). In particular, 5hmC patterns in promoter regions were able to divide the 54 AML patients in 3 classes using unsupervised hierarchical clustering.

Class 1 and 2 had similar clinical features whereas class 3 had clearly an enhanced blast count. Promoter 5hmC levels are reduced from class 1 to 3 in TET2 and enhanced in the WT gene. In cluster 3 more mutations in DNA methylation associated genes as TET2, IDH1, IDH2 and WT are found.  Also, overall survival and regions with differential 5hmC levels as compared to cluster 1 were assessed. Finally, enriched genes of these clusters were grouped according to their biological function using the IPA package from Qiagen.

The work is well performed and of interest since circulating DNA analysis is minimal invasive an may help to longitudinally monitor the disease and predict survival. However, sometimes the wording and description of the data is not suitable or highlights not the important facts. See my comments below.

1. The median overall survival given in part 3.3 of the study makes no sense given the low number of patients (below 15) in cluster 1 and 2. I would say that cluster 1 and 2 cannot be distinguished concerning overall survival (therefore stating that cluster 1 had a median survival of 8 month versus 22 months in cluster 2 makes no sense, despite it may be mathematically correct). You would need more patients and longer observation time to make this clear. 

The 12 month overall survival rate is here the better marker (as stated in the last sentence of page 4‼!)

2. Also Part 3.4 Genomic regions… is a bit obscure. It is not clear what we learn from the Venn diagram in Figure 4. As one idea can you compare your genes of interest that you show in Figure 4 to known RNA seq data in AML. That would be important to get an idea of functional meanings of differential 5hmC clusters.

3. Also Discussion of data from longitudinal assessment of 5hmC expression in a small group of patients would be of interest and comparison to the clusters found in this work.

4. In part 3.5 Signaling pathways. It is difficult to interpret 175 and 247 signalling pathways in clusters 2 and 3, respectively (page 6). Does this have any meaning?? I am not sure.

Better discuss the differences you found with PPARgamma in cluster 2 versus mTOR and ERK/MAPK in cluster 3, suggesting pro-proliferative signalling in the cluster with the worst prognosis.

5. The discussion is fine but the conclusions (part 5) you should really just state the most important points, which are the 3 clusters you defined (leave out the 4 lines discussing TET‼)

Say that cluster 1 and 2 have a similar prognosis but may be differential in the genetic makeup. This could mean that therapeutic approaches for cluster 1 and cluster 2 patients may differ (see PPARgamma). This is of interest‼

In cluster 3 it is clear that it has the worst prognosis and that mutations in DNA methylation genes are enriched in this cluster. What about multivariate analysis is cluster 3 a independent prognostic marker compared to blast count in the bone marrow?

I would put these points prominently into the conclusions.

Author Response

We are very grateful to the reviewers for their positive appraisal of our manuscript, and very much appreciate their thoughtful and constructive suggestions and comments. Following these suggestions, we have performed additional data analyses and have revised our manuscript carefully to address the reviewers’ concerns. All the text changes are tracked. We hope that this revised version of the manuscript will fulfill the requirements of the reviewers and editors, and that the manuscript will now be suitable for publication in Genes.

Reviewer 1.

Comments and Suggestions for Authors

In the manuscript by J. Shao et al. with the title “Classification of Acute Myeloid Leukemia by Cell-Free DNA 5-hydroxymethlcytosine” the authors describe the use of cell free circulating DNA to classify AML using the epigenetic mark 5-hydroxymethlcytosine (5hmC). The letter is the first step of the demethylation of 5-methlycytosine by TET proteins. The authors perform the analysis in a genome wide fashion and then restrict their analysis to different regions of the genome defined by specific markers on histone tails, that preferentially bind promoters (H3K4me3), cis-regulatory regions (H3K27ac, H3K4me) or gene bodies H3K36me). In particular, 5hmC patterns in promoter regions were able to divide the 54 AML patients in 3 classes using unsupervised hierarchical clustering.

Class 1 and 2 had similar clinical features whereas class 3 had clearly an enhanced blast count. Promoter 5hmC levels are reduced from class 1 to 3 in TET2 and enhanced in the WT gene. In cluster 3 more mutations in DNA methylation associated genes as TET2, IDH1, IDH2 and WT are found.  Also, overall survival and regions with differential 5hmC levels as compared to cluster 1 were assessed. Finally, enriched genes of these clusters were grouped according to their biological function using the IPA package from Qiagen.

The work is well performed and of interest since circulating DNA analysis is minimal invasive an may help to longitudinally monitor the disease and predict survival. However, sometimes the wording and description of the data is not suitable or highlights not the important facts. See my comments below.

Reviewer’s comment 1: The median overall survival given in part 3.3 of the study makes no sense given the low number of patients (below 15) in cluster 1 and 2. I would say that cluster 1 and 2 cannot be distinguished concerning overall survival (therefore stating that cluster 1 had a median survival of 8 month versus 22 months in cluster 2 makes no sense, despite it may be mathematically correct). You would need more patients and longer observation time to make this clear. 

The 12 month overall survival rate is here the better marker (as stated in the last sentence of page 4‼!)

Response: We agree with reviewer’s comment and removed the sentence “The median OS was 8.9 months (range 4.4 – 29.8 months) for cluster 1, 22.1 months (range 1.7 – 45.2 months) for cluster 2, and 6.8 months (range 0.2 – 81.8 months) for cluster 3” from the main text. Please note, the number of patients was listed below the survival curve in Figure 3 and did exceed 15.

Reviewer’s comment 2: Also Part 3.4 Genomic regions… is a bit obscure. It is not clear what we learn from the Venn diagram in Figure 4. As one idea can you compare your genes of interest that you show in Figure 4 to known RNA seq data in AML. That would be important to get an idea of functional meanings of differential 5hmC clusters.

Response: We compared our gene list (DhMR-linked genes) with RNA seq data in AML patients from Gene Expression Profiling Interactive Analysis (GEPIA) database. The majority of DhMR-linked genes were differentially expressed at mRNA level. Please see Figure 4 and lines 225-232.

Reviewer’s comment 3: Also Discussion of data from longitudinal assessment of 5hmC expression in a small group of patients would be of interest and comparison to the clusters found in this work.

Response: We agree that it would be of interest to discuss data from a longitudinal assessment of 5hmC changes. Unfortunately, we do not have longitudinal samples for the assessment. This could be a project we will work on in the future.

Reviewer’s comment 4: In part 3.5 Signaling pathways. It is difficult to interpret 175 and 247 signalling pathways in clusters 2 and 3, respectively (page 6). Does this have any meaning?? I am not sure.

Better discuss the differences you found with PPARgamma in cluster 2 versus mTOR and ERK/MAPK in cluster 3, suggesting pro-proliferative signalling in the cluster with the worst prognosis.

Response: We have revised the text to emphasize the differences between cluster 2 and 3 and removed the sentence “These genes were significantly enriched in 175 signaling pathways in cluster 2 and 247 signaling pathways in cluster 3” from the main text. We also revised the descriptions for specific signaling pathways for cluster 2 and 3. Please see text lines 251-257.

Reviewer’s comment 5: The discussion is fine but the conclusions (part 5) you should really just state the most important points, which are the 3 clusters you defined (leave out the 4 lines discussing TET‼)

Say that cluster 1 and 2 have a similar prognosis but may be differential in the genetic makeup. This could mean that therapeutic approaches for cluster 1 and cluster 2 patients may differ (see PPARgamma). This is of interest‼

In cluster 3 it is clear that it has the worst prognosis and that mutations in DNA methylation genes are enriched in this cluster. What about multivariate analysis is cluster 3 a independent prognostic marker compared to blast count in the bone marrow?

I would put these points prominently into the conclusions.

Cluster 3 is independent prognostic marker

Response: We have removed the four lines discussing TET and added therapeutic indications for cluster 2. Multivariate cox regression analysis shows that cluster 3 is an independent prognostic marker compared to blast count in the bone marrow. Please see lines 344-353 in the main text.

Reviewer 2 Report

The manuscript under review presents a novel approach for stratifying AML patients into risk groups based on 5-hydroxymethylcytosine (5hmC) gene profiling in ctDNA. The manuscript is scientifically sound and methodologically reasonable.  There are no major comments to the manuscript. Minor points:

1 I would suggest to add more data concerning ctDNA features. What was the yield of ctDNA? What was the variance in yields between the patients? Between clusters of patients?   Same applies to the the overall length of ctDNA fragments and its difference (if any) between patients and patients cohorts.  These details seem important, because at the moment the methods of ctDNA isolation and analyzing are not sufficiently standardized and a more detailed description can help other researchers reproduce the results obtained by the authors. 

2 It is well expected, that high-risk AML patients have more complications like infections, inflamations, etc. due to more aggressive treatment and severe neutropenia. These conditions are known  to increase the levels of free circulating DNA (non-tumor). One can speculate that the excess of this free non-tumor DNA in plasma may bias the measurement of 5hmC in circulating tumor DNA. How do the authors challenge this assumption?

Author Response

We are very grateful to the reviewers for their positive appraisal of our manuscript, and very much appreciate their thoughtful and constructive suggestions and comments. Following these suggestions, we have performed additional data analyses and have revised our manuscript carefully to address the reviewers’ concerns. All the text changes are tracked. We hope that this revised version of the manuscript will fulfill the requirements of the reviewers and editors, and that the manuscript will now be suitable for publication in Genes.

Reviewer 2.

The manuscript under review presents a novel approach for stratifying AML patients into risk groups based on 5-hydroxymethylcytosine (5hmC) gene profiling in ctDNA. The manuscript is scientifically sound and methodologically reasonable.  There are no major comments to the manuscript. Minor points:

Reviewer’s comment 1: I would suggest to add more data concerning ctDNA features. What was the yield of ctDNA? What was the variance in yields between the patients? Between clusters of patients?   Same applies to the the overall length of ctDNA fragments and its difference (if any) between patients and patients cohorts.  These details seem important, because at the moment the methods of ctDNA isolation and analyzing are not sufficiently standardized and a more detailed description can help other researchers reproduce the results obtained by the authors. 

Response: We analyzed cfDNA characteristics in all AML patient samples and compared different clusters. We have revised the text to detail our methods for assessing cfDNA quantity and quality and cfDNA characteristics in the Results section. Please see lines 92-95 and 163-173 in the main text.

Reviewer’s comment 2: It is well expected, that high-risk AML patients have more complications like infections, inflamations, etc. due to more aggressive treatment and severe neutropenia. These conditions are known  to increase the levels of free circulating DNA (non-tumor). One can speculate that the excess of this free non-tumor DNA in plasma may bias the measurement of 5hmC in circulating tumor DNA. How do the authors challenge this assumption?

Response: We agree that infections, inflammations, and other factors may influence the constitution of cfDNA. However, 5hmC, like most other epigenetic markers, is highly tissue and cancer-type specific (PMID: 34253716, 33268789, 27477909, 30922396, 34957093). Therefore, it is unlikely that cfDNA derived from sources other than cancer cells would significantly affect our 5hmC analysis. Specifically, we expect that 5hmC patterns in AML markedly differ from those associated with infections and inflammations. Supporting this, cfDNA 5hmC has been successfully used as a sensitive marker for disease diagnosis, prognosis, and treatment response prediction in a variety of diseases, including cancers (PMID: 30922396, 34957093). We have added comments in the Discussion lines 332-334.

Round 2

Reviewer 1 Report

I agree to the new version of the manuscript and have no more comments.

The comparison of the mRNAs in the Venn diagram now makes it more meaningful